# Antibacterial Activity and Cytocompatibility of Electrospun PLGA Scaffolds Surface-Modified by Pulsed DC Magnetron Co-Sputtering of Copper and Titanium

**DOI:** 10.3390/pharmaceutics15030939

**Published:** 2023-03-14

**Authors:** Arsalan D. Badaraev, Marat I. Lerner, Olga V. Bakina, Dmitrii V. Sidelev, Tuan-Hoang Tran, Maksim G. Krinitcyn, Anna B. Malashicheva, Elena G. Cherempey, Galina B. Slepchenko, Anna I. Kozelskaya, Sven Rutkowski, Sergei I. Tverdokhlebov

**Affiliations:** 1Weinberg Research Center, School of Nuclear Science & Engineering, National Research Tomsk Polytechnic University, 30 Lenin Avenue, 634050 Tomsk, Russia; 2Institute of Strength Physics and Materials Sciences of Siberian Branch of the Russian Academy of Sciences, 2/4 Akademicheskii Avenue, 634055 Tomsk, Russia; 3Institute of Cytology RAS, 4 Tikhoretsky Avenue, 194064 Saint Petersburg, Russia

**Keywords:** electrospinning, PLGA scaffolds, pulsed DC magnetron co-sputtering, copper-titanium thin film, surface modification, antibacterial activity, cytotoxicity

## Abstract

Biocompatible poly(lactide-co-glycolide) scaffolds fabricated via electrospinning are having promising properties as implants for the regeneration of fast-growing tissues, which are able to degrade in the body. The hereby-presented research work investigates the surface modification of these scaffolds in order to improve antibacterial properties of this type of scaffolds, as it can increase their application possibilities in medicine. Therefore, the scaffolds were surface-modified by means of pulsed direct current magnetron co-sputtering of copper and titanium targets in an inert atmosphere of argon. In order to obtain different amounts of copper and titanium in the resulting coatings, three different surface-modified scaffold samples were produced by changing the magnetron sputtering process parameters. The success of the antibacterial properties’ improvement was tested with the methicillin-resistant bacterium Staphylococcus aureus. In addition, the resulting cell toxicity of the surface modification by copper and titanium was examined using mouse embryonic and human gingival fibroblasts. As a result, the scaffold samples surface-modified with the highest copper to titanium ratio show the best antibacterial properties and no toxicity against mouse fibroblasts, but have a toxic effect to human gingival fibroblasts. The scaffold samples with the lowest copper to titanium ratio display no antibacterial effect and toxicity. The optimal poly(lactide-co-glycolide) scaffold sample is surface-modified with a medium ratio of copper and titanium that has antibacterial properties and is non-toxic to both cell cultures.

## 1. Introduction

Biodegradable and biocompatible synthetic polymers have found numerous applications in tissue engineering, such as the regeneration of neural [1], bone [2] and soft tissues [3]. The following are used the most under all synthetic polymers: polycaprolactone (PCL), polylactide (PLA) and the copolymer poly(lactide-co-glycolide) (PLGA) [4]. PCL possess good mechanical properties (mechanical resistance, deformability) [5], but has very slow degradation rate (up to several years in vivo) [6]. PLA has a high compressive strength [7], but has a degradation rate in the range of months [8]. Among the most common natural polymers used in tissue engineering are collagen and gelatin [9]. Despite the high biocompatibility of collagen and gelatin, they have low mechanical strength [10,11] and a very high rate of degradation. The degradation of collagen is up to 4 weeks in vivo and a complete degradation of gelatin up to 2–4 weeks [12,13]. Compared to PCL, PLA, collagen and gelatin, PLGA has a controlled degradation rate that depends on the ratio of lactide/glycolide for synthesis [14]. PLGA has been approved by the US Food and Drug Administration for clinical use [15]. This happened because PLGA has good biocompatibility and mechanical properties that are suitable for applications in tissue engineering [16]. PLGA is also of high interest because of its use for drug delivery applications from a biodegradable polymer matrix [17].

Electrospinning is commonly used method for the fabrication of non-woven highly porous, biocompatible and mechanically stable polymeric scaffolds, which consist from chaotically interlaced fibers with diameters from several nanometers to microns [18]. It is possible to fabricate a polymer scaffold with a structure that mimics the topology of extracellular matrix by electrospinning [4]. An extracellular matrix structure is optimal for cell cultivation and differentiation, therefore attempts are being made to produce biocompatible materials whose design is most similar to such a matrix [19,20]. PLGA scaffolds fabricated by electrospinning are actively used in biomedicine for the regeneration of neural [1], periodontal [21], bone [22] and connective [23] tissues.

Despite the advantages mentioned, PLGA has no appreciable antibacterial properties. The surgical insertion of materials without antibacterial properties into a living body increases the risk of a bacterial contamination, which can lead to dangerous postoperative complications. A number of studies are dedicated to the creation of polymer scaffolds with antibacterial properties obtained by incorporation of antibiotics [24,25,26]. The excessive use of antibiotics over decades has led to the formation of antibiotic-resistant pathogens [27]. In the last decade, the number of bacterial strains that have become multi-resistant to several types of antibiotics has increased significantly [28]. The search for solutions against antibiotic-resistant bacteria is currently one of the great challenges in medicine [29].

Metals such as silver, copper and zinc have been successfully used against antibiotic-resistant bacteria [30,31,32,33]. Among metals with antibacterial properties, one of the most promising is copper. Numerous studies have shown that copper-polymer composites, copper nanoparticles, and copper-containing thin films have antimicrobial properties [34,35,36,37]. Due to its antibacterial properties, copper prevents the formation of biofilms [38,39]. Compared to the popular silver, copper is much cheaper and compared to zinc, copper has stronger antibacterial properties [40].

Despite its ability to defang antibiotic-resistant bacteria, copper is toxic, like other metals with this property [41]. In order to overcome the toxicity of copper, thin films of copper-titanium (Cu-Ti) alloys are formed, which have both good antibacterial and biocompatible properties [42,43]. Magnetron sputtering is one of the best-known methods for producing composite thin films [44,45]. Direct current (DC) magnetron co-sputtering in pulsed mode allow to fabricate Cu-Ti alloy thin films with high purity, homogeneity and good mechanical properties (hardness, wear resistance) [46]. Compared to micro-arc oxidation and chemical vapor deposition, pulsed DC magnetron sputtering can be used to modify the surface of temperature-sensitive biodegradable polymers. The modification of polymer scaffolds by magnetron sputtering allows to preserve their morphology, structure and mechanical properties [47].

Pulsed DC magnetron co-sputtering of the targets Cu and Ti allows PLGA scaffolds to possess antibacterial properties against methicillin-resistant Staphylococcus aureus (MRSA) and retain the original biocompatibility as an unmodified PLGA scaffold. The aim of the present work is to investigate the influence of Cu-Ti coatings on antibacterial properties and cell toxicity of electrospun PLGA scaffolds.

## 2. Materials and Methods

A schematic overview of the PLGA scaffold fabrication via electrospinning and their further surface modification by magnetron co-sputtering, as well as all performed investigation methods is shown in Figure 1.

### 2.1. Materials

Poly(lactide-co-glycolide) (PLGA) with a copolymer ratio of 85/15 (PLA/PGA), with the molecular weights of M_n_ ≈ 202,000 g/mol and M_w_ ≈ 338,000 g/mol, was purchased from Corbion Purac (Amsterdam, The Netherlands). Hexafluoroisopropanol ((CF_3_)_2_CHOH) with a purity of 99% was purchased from P&M Invest (Moscow, Russia).

Methicillin-resistant Staphylococcus aureus (MRSA strain AATCC 43300) has been obtained from BioVitrum (Novosibirsk, Russia). NIH/3T3 (ATCC CRL-1658) fibroblast cell lines were obtained from the State Research Center of Virology and Biotechnology (VECTOR, Novosibirsk, Russia). Primary cell cultures of human gingival fibroblasts (HGFs) were obtained from Pokrovsky Stem Cell Bank (Saint Petersburg, Russia).

### 2.2. Fabrication of Poly(lactide-co-glycolide) (PLGA) Scaffolds

A 4 wt% solution of PLGA was prepared by dissolving PLGA granules in hexafluoroisopropanol. Using an electrospinning setup (NANON-01A, MECC, Fukuoka, Japan) equipped with a cylindrical collector (length: 200 mm, diameter: 100 mm), the PLGA solution (placed in a syringe with a volume of 10 mL) was electrospun to obtain polymer scaffolds. The following electrospinning process parameters were used to fabricate the polymer scaffolds: - nozzle voltage: +22 kV, - distance between the nozzle and the collector: 150 mm, - electrospinning solution flow rate: 4 mL∙h^−1^, - spinneret width: 200 mm, - 20 G needle, - collector rotation speed: 200 rpm. During the electrospinning of the scaffold samples, the temperature in the electrospinning setup was in the range of 26 ± 4 °C and the relative humidity was in the range of 30 ± 18%.

### 2.3. Surface Modification

The surface modification of the PLGA scaffolds was carried out by pulsed DC magnetron co-sputtering of copper (Cu, 99.95%) and titanium (Ti, 99.95%) targets in an argon (Ar, 99.99%) atmosphere using the setup described in reference [48]. The targets had a disk-like shape with a diameter of 90 mm and a thickness of 8 mm. Two separate power supplies (APEL-M-5PDC, Applied Electronics, Tomsk, Russia) were used to power the first magnetron with the copper target and the second magnetron with the titanium target operating in a unipolar mode (according to reference [49]) with a pulse frequency of 100 kHz and a duty cycle of 70%.

Applied operating parameters of the surface modification of the PLGA scaffolds by magnetron co-sputtering are presented in supporting information (Appendix A). The unmodified PLGA scaffold will be marked with the letter “u”, the surface modified PLGA scaffolds will be marked as: “LCu-Ti”—for a low concentration of Cu in the Cu-Ti alloy coating, “MCu-Ti”—for the medium concentration of Cu and Ti in the Cu-Ti alloy coating, and “HCu-Ti”—for a high concentration of Cu in the Cu-Ti alloy coating. Note, one PLGA scaffold was only surface-modified with Ti to compare this type of coating with the Cu-Ti alloy coating types.

Halfway through the time of magnetron sputtering of the PLGA scaffold surfaces, there was a 20 min break, after which the process continued to completion. This break is required during surface modification by magnetron sputtering to avoid damage to the PLGA scaffolds that could be caused by the high temperature of the modification process. In the sputtering chamber, the PLGA scaffold samples were rotated in an axial alignment with a rotation radius of 150 mm.

The resulting deposition rates of the thin film coatings fabricated by the applied operating parameters (Appendix A) were determined using a quartz thickness gauge (Micron-5, Izovac, Minsk, Belarus): LCu-Ti—7.5 nm∙min^−1^, MCu-Ti—6.8 nm∙min^−1^ and HCu-Ti—10.3 nm∙min^−1^. Subsequently, the required magnetron sputtering times were calculated using these deposition rates in order to obtain coating thicknesses of approximately 200 ± 10 nm.

### 2.4. Scanning Electron Microscopy

To investigate the morphology of PLGA scaffolds, micrographs were taken at a 500× magnification using a scanning electron microscope (SEM, JCM-6000 Plus, Jeol, Akishima, Japan) at an accelerating voltage of 15 kV. SEM micrographs at 15,000× magnification were obtained with a LEO EVO 50 (Carl Zeiss AG, Oberkochen, Germany) at an accelerating voltage of 20 kV. In order to dissipate the accumulated charge, a thin film of gold was deposited on surface of the PLGA scaffolds using a sputter coater (Smart Coater, Jeol Ltd., Akishima, Japan).

PLGA scaffold fiber diameters and pore area histograms were calculated using SEM micrographs taken at 500× magnification and at an accelerating voltage of 15 kV with the DiameterJ v1.018 plug-in (National Institute of Standards and Technology, Gaithersburg, MD, USA) for Fiji/ImageJ image processing software (National Institute of Health, Washington, DC, USA).

### 2.5. Porosity of the Scaffolds

The porosity values (*P*) of the PLGA scaffolds were determined using the gravimetric method [45] and calculated according the following equation [50]:(1)$$P=1-\rho_{scaffold}/\rho_{solid}$$
with: $\rho_{\mathrm{scaffold}}$—the density of porous scaffolds in g∙cm^−3^, $\rho_{\mathrm{solid}}$—the density of bulk PLGA: 1.25 g∙cm^−3^ [51].

### 2.6. Atomic Force Microscopy

Morphology images and roughness of unmodified and modified PLGA scaffold surfaces were obtained by atomic force microscopy (AFM, NTEGRA NT-MDT AFM system, NT-MDT Spectrum Instruments, Moscow, Russia) in the semi-contact mode. Samples were scanned using monocrystalline silicon cantilevers NSG01 (NT-MDT Spectrum Instruments, Moscow, Russia) with a force constant of 1.45–15.10 N∙m^−1^. AFM micrographs were obtained over scanned areas of 40 × 40 µm^2^ and 1 × 1 µm^2^. The scanned area 40 × 40 µm^2^ was used for investigation of micromorphology and roughness fibrous structure, the scanned area 1 × 1 µm^2^ was used for analysis of nanomorphology and roughness of distinct fiber surfaces. Subsequently, the mean surface roughness for each PLGA scaffold sample was calculated from 3–5 AFM micrographs.

### 2.7. Elemental and Chemical Analysis

Via energy dispersive X-ray spectroscopy (EDX), elemental compositions of the produced scaffolds were investigated with an EDX analyzer (INCA, Carl Zeiss, Oberkochen, Germany) at an accelerating voltage of 20 kV, which is connected to the SEM LEO EVO 50. Thereafter, the obtained elemental composition data was corrected by the quantitative analysis method ZAF (short for: Z—atomic number, A—absorption effect, F—fluorescence excitation effect). The analysis was provided at a magnification of 500× and the analyzed area was round about 0.25 mm^2^.

Raman spectra were recorded by NTEGRA NT-MDT AFM-Raman system (NT-MDT Spectrum Instruments, Moscow, Russia), equipped with a green laser (λ = 532 nm) at a magnification of 100×.

X-ray photoelectron spectroscopy (XPS) was performed for the chemical analysis of unmodified and Cu-Ti modified PLGA scaffolds using a surface analysis system (Nexsa XPS system, Thermo Fisher Scientific Inc., Waltham, MA, USA). For the XPS investigations of the observed PLGA scaffold surfaces, the following parameters were applied: Al Kα radiation source with a photon energy of 1486.6 eV, pass energy for survey and core-level spectra of 200 eV and 50 eV, spot size of the X-ray beam of 400 µm. Deconvolution of core-level spectra were carried out with Casa XPS software (Thermo Fisher Scientific Inc., Waltham, MA, USA) after performing a Shirley background correction. XPS peaks were fitted via Voigt function with 80% Gaussian and 20% Lorentzian character.

### 2.8. Morphology and Elemental Mapping of Sample Cross-Sections

SEM cross-section micrographs at a magnification of 2500× and EDX mapping of titanium (Ti), copper (Cu), carbon (C) and oxygen (O) were taken with an energy dispersive analyzer (INCA, Carl Zeiss AG, Oberkochen, Germany) at an accelerating voltage of 20 kV and attached to the LEO EVO 50 (Carl Zeiss AG, Oberkochen, Germany). The analyzed sample area was ~0.06 mm^2^.

### 2.9. Mechanical Properties

A tensile testing machine (Instron 3343, Illinois Tool Works, Glenview, IL, USA) was utilized to measure the mechanical characteristics of the PLGA scaffold samples. This tensile testing machine was equipped with a 50 N static load cell (Instron 2519-102, Illinois Tool Works, Glenview, IL, USA) and run at a traverse speed of 10 mm∙min^−1^. The tested PLGA scaffold sample size between the traverse jaws of the tensile tester was 10 × 10 mm^2^. For these measurements, three PLGA samples of each sample type were used.

### 2.10. Wettability

Water contact angles (WCA) of the fabricated PLGA scaffolds were determined by a drop shape analyzer (EasyDrop DSA-20, KRÜSS, Hamburg, Germany) using the sessile drop method. Three droplets of deionized water with a volume of 2 µL were applied on each sample in air. Micrographs of the water droplets were taken one minute after application to the PLGA sample surfaces to determine the water contact angles.

### 2.11. Thermal Gravimetric Analysis

Thermal gravimetric analysis (TGA) was performed using an SDT-Q600 (TA Instruments, New Castle, DE, USA) to investigate the temperature-dependent mass change of the polymer scaffold samples. For this purpose, the PLGA scaffolds were cut into square samples with an area of 1 × 1 cm^2^ with a weight in the range of 3–4 mg. The prepared samples were measured in heat-resistant pot in a temperature range from 20 °C to 800 °C at a heating rate of 20 °C∙min^−1^ under air atmosphere.

### 2.12. Copper Ions Release 

Concentrations of copper ions were measured by stripping voltammetry. For this purpose, the Cu-Ti modified PLGA scaffolds with an area of 0.5 cm^2^ were loaded into tubes filled with 1 mL of deionized water (Milli-Q^®^, Merck Millipore, Burlington, MA, USA) with a conductivity of 1 μS∙cm^−1^. Thereafter, the tubes were placed in an incubation chamber at 37 °C. After 0 (30 min), 1, 3, 7, 14 and 28 days, the modified PLGA samples were removed from the tubes with deionized water. Thereafter, the samples were analyzed in a two-electrode measurement system. Voltammograms were recorded at a linear potential sweep in differential mode using a STA voltammetric system (TU no. 4215-001-20694097-98, ITM, Tomsk, Russia). The copper ion release rates (RR) were calculated according the following equation [52]:(2)$$RR=(C_{t+\Delta t}-C_{t})/\left(\Delta t\right)$$
with: $C_{t}$—concentration of copper ions in deionized water on at time “*t*” in mg∙L^−1^, $C_{t+\Delta t}$—concentration of copper ions at “*t* + ∆*t*”, ∆*t*—the difference time for the interaction of the samples with water, at which two different concentrations were observed ($C_{t+\Delta t}$ and $C_{t}$) in days (d). Copper ion release rates were calculated in a similar way in other studies [53,54].

A mercury-film electrode with a silver rod (length: 10 mm, diameter: 1 mm) was used as indicator electrodes. Mercury solution of 300 mg∙L^−1^ was used as a modifier.

A silver chloride electrode (SCE) filled with a 1 mol∙L^−1^ potassium chloride solution was used as reference electrode. Therefore, the electrode was filled with a solution of potassium chloride with a concentration of 1 mol∙L^−1^, the hole was left plugged for at least 2 h to reach the equilibrium potential during the first filling. After completion of the analysis, the electrode was immersed in a unipolar potassium chloride solution.

### 2.13. Morphology and Elemental Mapping of the Sample Surfaces

For the investigation of the morphology and the elemental mapping of PLGA scaffolds, the samples were prepared in the same way as described in Section 2.12: PLGA scaffold samples with a surface area of 0.5 cm^2^ were placed into tubes filled with 1 mL of deionized water. Three groups of samples were studied: the first group were not placed in deionized water (initial, before interaction), the second group were removed from the test tubes after 14 days and the third group were removed from test tubes after 28 days. In order to prepare the samples for the morphology study via scanning electron microscopy (SEM) and the elemental mapping via energy-dispersive X-ray spectroscopy (EDX), the samples were dried in a vacuum oven at room temperature and a pressure of 100 Pa for 48 h. SEM and EDX mapping micrographs of the PLGA scaffold samples were received at a magnification of 5000× and an accelerating voltage of 30 kV using a Quanta 200 (FEI Company, Hillsborough, OR, USA).

### 2.14. Antimicrobial Activity

To determine the antibacterial activities of the PLGA scaffold samples, the antimicrobial fabric test ISO 20743:2013 was applied. In a sterile tube containing a PLGA scaffold sample with an area of 1.0 cm^2^, 0.2 mL MRSA bacterial suspension with a concentration of 10^5^ CFU∙mL^−1^ (CFU—colony-forming unit) was added. Thereafter, the tubes containing the PLGA scaffold samples in the bacteria suspension were incubated at 37 °C for 6, 12 and 24 h. After incubation, 2 mL of sodium chloride solution (NaCl 0.85% in aqueous solution, Groteks, St. Petersburg, Russia) was added to the tubes and stirred using a vortex mixer (Heidolph, Schwabach, Germany). After mixing, the solution in tubes was plated on Petri dishes containing Mueller–Hinton agar (MHA, National Research Center of Pharmacotherapy, St. Petersburg, Russia). Subsequently, the MHA Petri dishes were incubated at 37 °C for 24 h, after which the bacteria colonies that had grown were manually counted. A control group were treated in the same way, but in the absence of PLGA scaffolds. Three PLGA samples were measured for each sample type.

The antibacterial activities *R* were calculated according to the following equation [55]:(3)R=100%·((C−A)/C)
where *A* is the number of bacteria counted from the MHA Petri dishes with the bacteria suspension obtained treated with a PLGA scaffold, *C* is the number of bacteria counted from the control group MHA Petri dishes, incubated for 24 h and without PLGA scaffolds.

### 2.15. Cytotoxicity towards NIH/3T3 Cells

The NIH/3T3 cells were grown as a monolayer in minimum essential medium (MEM, Lonza, Basel, Swiss) with addition of 10% fetal bovine serum (HyClone, Logan, UT, USA), 2 mM L-glutamine (HyClone, Logan, UT, USA) and 1% penicillin/streptomycin (HyClone, Logan, UT, USA). The cell cultivation was performed at 37 °C and 5% CO_2_ within 24 h. MEM media was used as a model biological fluid for the biodegradation study of the samples according to ISO 10993-5. The PLGA samples were extracted for 24 h at 37 °C at a surface-to-volume ratio of 1 cm^2^/mL MEM medium. A sample of the medium was incubated under similar conditions and was used as a negative control. All experiments were repeated in triplicates.

MTT test is based on the reaction of 3-(4,5-dimethylthiazol-2-yl)-2,5-tetrazolium bromide (MTT; Paneco, Moscow, Russia) with reductases in living cells, whereby formazan is formed, which stains the extract purple. Cells (10^4^ cells in 100 μL MEM) were placed in each well of a sterile 96-well microplate and incubated in a humidified atmosphere of 5% CO_2_ at 37 °C to reach approximately 90% confluence. Subsequently, 100 μL of a sample extract solution was added and incubated for 24 h, 48 h and 96 h at 37 °C and 5% CO_2_. A TC20 Automated Cell Counter (Bio-Rad, Moscow, Russia) was used for cell counting. Subsequently, the MTT solution was added for 2 h at 37 °C and 5% CO_2_. Thereafter, the optical densities (OD) were determined using a microplate spectrophotometer Thermo Scientific Multiskan FC (Thermo Fisher Scientific, Waltham, MA, USA) at a wavelength of 570 nm. The relative growth rate (RGR) was calculated based on the following equation [56]:(4)$$RGR=100\%\cdot \left(OD_{sample}/OD_{blank}\right)$$
where *OD_sample_* is optical density of cells incubated with Cu-Ti modified PLGA scaffolds, *OD_blank_* is optical density of control test tubes (medium without samples). Micrographs of NIH/3T3 cells were taken using an optical microscope at 500× magnification (Axio Vert A1 Mat, Carl Zeiss AG, Oberkochen, Germany).

### 2.16. Cytotoxicity towards Human Gingival Fibroblasts

Cytotoxic properties of the PLGA scaffold samples were tested using 3-(4,5-dimethylthiazol-2-yl)-2,5-diphenyltetrazolium bromide (MTT, Thermo Fisher Scientific, Waltham, MA, USA) assay. PLGA scaffold samples were placed in 24-well plates, then human gingival fibroblasts (HGFs) were added at a density of 40,000 cells per well. All experiments were repeated in triplicates.

Wells without samples were used as controls. After 24 h of cultivation, the MTT assay was stopped. The optical density (OD) of the cells stained with MTT was measured using a Varioskan LUX Multimode Microplate Reader (Thermo Scientific, Waltham, MA, USA) at a wavelength of 590 nm. The relative growth rates (RGR) were calculated according to Equation (4).

### 2.17. Statistics

Statistical data processing was performed using the OriginPro^®^ 2021 program (OriginLab, Northampton, MA, USA) and STATISTICA 10.0 (StatSoft, Tulsa, OK, USA). Differences in fiber diameters, pore area, mechanical properties, wettability, antibacterial properties and toxicity of PLGA scaffolds were evaluated using the Mann–Whitney U test. The differences were statistically significant at *p* < 0.05.

## 3. Results and Discussion

### 3.1. Morphology of the PLGA Scaffold Surfaces

SEM images at 500× and 15,000× magnification and AFM images of the surface-modified PLGA scaffolds are shown in Figure 2.

SEM micrographs at 500× magnification and AFM micrographs at 40 × 40 μm^2^ scanning area show that the PLGA scaffolds were composed of randomly intertwined fibers (Figure 2, the first and third column). Mean fiber diameter of unmodified and surface-modified PLGA scaffold samples ranged from (1.29–1.40) ± 0.59 μm (Appendix A, left column). Moreover, mean pore areas were in the range of (15.7–18.9) ± 14.8 μm^2^, with a porosity in the range of (83.2–86.4) ± 4.5% (Appendix A, right column). Surface roughness has been measured at 40 × 40 μm^2^ scanning area and was for unmodified PLGA scaffolds in the range of 756 ± 38 nm (Figure 2, the third column). The surface roughness of surface-modified PLGA scaffolds was in the range of (680–747) ± 63 nm (Figure 2, the third column). As we can see, the surface modification of PLGA scaffolds by the selected process parameters of magnetron sputtering practically did not significantly change the mean fiber diameter, mean pore area and porosity (Appendix A), due to a low amount of deposited coating material in comparison with fiber diameter, as the surface roughness measured at 40 × 40 μm^2^ scanning area (Figure 2, third column) confirmed. It is statistically confirmed (*p* < 0.05) that there are no significant differences in terms of average fiber diameter, pore area, and porosity between unmodified and surface-modified PLGA scaffolds.

Based on the SEM images at 15,000× magnification and AFM images at 1 × 1 μm^2^ scanning area was demonstrated, that unmodified PLGA scaffolds fibers have a smooth surface (Figure 2, second and fourth column). The Cu-Ti composite thin films on the PLGA scaffold samples led to the formation of a more distinct morphology and thus the appearance of a rougher structure on the surface of individual fibers (Figure 2, second and fourth column). Separate spherical clusters could be observed on the surface of modified fibers of PLGA scaffolds (Figure 2, fourth column). The distinct fibers of the unmodified PLGA scaffold samples had a relatively low surface roughness of 4.6 ± 3.1 nm (Figure 2, fourth column). In contrast, the fibers of LCu-Ti and MCu-Ti PLGA scaffolds had a surface roughness of 6.0 ± 3.7 nm and 5.6 ± 2.9 nm (Figure 2, fourth column), respectively. The fibers of HCu-Ti scaffolds showed the highest roughness values in the range of 8.8 ± 4.8 nm (Figure 2, fourth column). After surface-modification, an increase in roughness and the appearance of spherical clusters on the surface of individual fibers were observed, which indicated the formation of copper-titanium thin films on PLGA scaffold fibers (Figure 2, fourth column). The HCu-Ti PLGA samples had the highest roughness and highest size of spherical clusters, which was associated with the modification mode, characterized by the highest discharge power for the Cu target (Appendix A). An increase in the discharge power led to an increase in the deposition rate, which increased the roughness and size of clusters on the surface of deposited metal thin films [57].

Thus, the modification process did not significantly affect the fibrous surface morphology of PLGA scaffolds (Figure 2, first and third column). However, the surface-modification increased the roughness and promotes the formation of spherical clusters on the surface of individual PLGA fibers (Figure 2, second and fourth column).

### 3.2. Elemental and Chemical Composition of the PLGA Scaffold Surfaces

The EDX spectra of unmodified PLGA scaffolds demonstrated that the scaffold surfaces composed from carbon (C) and oxygen (O) that forming the polymer backbone (Appendix A). EDX analysis revealed that the elements copper (Cu) and titanium (Ti) were detected after surface modification of the PLGA scaffolds, indicating the presence of a composite thin film consisting of these elements. The C/O ratio of unmodified and modified PLGA scaffolds varied within a small range (1.00–1.22). The total atomic concentration of the elements Cu and Ti for the surface-modified PLGA scaffolds ranged from 15.0% to 15.5%, indicating a comparable atomic concentration in all surface-modified scaffold samples. Moreover, the LCu-Ti PLGA scaffold samples had the lowest Cu/Ti ratio (1.08) and the HCu-Ti samples the biggest (2.33).

The XPS survey spectra of unmodified and surface-modified PLGA scaffolds are presented in the supporting information (Appendix A). Two peaks, one for carbon (C1s) and one for oxygen (O1s), were present in the XPS spectrum of the unmodified PLGA scaffold sample. For the unmodified PLGA sample, the C/O ratio was 1.98. After PLGA scaffolds modification, C/O ratio began to change in a wide range of 0.77–1.63. After surface modification, copper (Cu) and titanium (Ti) were present on the upper side of the PLGA scaffold fibers. The surface-modified PLGA sample LCu-Ti had the lowest Cu/Ti ratio (0.43), the HCu-Ti samples exhibited the highest Cu/Ti ratio (1.37).

As can be seen in the Raman spectrum of the unmodified PLGA scaffold, three high intensity peaks at the Raman shift of 877, 1458, and 1775 cm^−1^ were present (Appendix A). These Raman peaks were typical for nonwoven PLGA scaffolds [58]. After Cu-Ti modification of the scaffolds, the PLGA related peaks were no longer observed in the Raman spectra, which may indicate the formation of a copper-titanium thin film that completely covers the PLGA sample fibers (Appendix A). The obtained Raman spectrum for the Cu-Ti modified PLGA sample MCu-Ti correlated with studies of copper oxide thin films [59] and copper-titanium oxide thin films [60], deposited by magnetron sputtering. After amplification of the laser beam energy, in the Raman spectrum of the MCu-Ti sample, a broad peak in the region of 1200 to 1600 cm^−1^ was observed, which indicated the formation of amorphous carbon [61], which could have been formed due to melting of the temperature-sensitive PLGA.

While the Raman spectra of the modified PLGA sample exhibited no peaks (Appendix A), the XPS spectra of the modified samples show carbon peaks C1s with concentrations ranging from 35.2 to 53.3% (Appendix A). The Raman spectra of unmodified PLGA scaffolds did not show peaks because the metal coating on the fibers can easily reflect the light, which significantly reduces the signal intensity. In addition, the absence of polymer peaks in the Raman spectra and the significant change in the intensity of C1s peaks in the XPS spectra of the surface-modified PLGA scaffolds might be related to carbon contamination [62]. According to reference [62], after argon ion beam etching of surface-modified PLGA scaffolds, the intensity of C1s spectra decreased and the intensity of Ti2p and O1s spectra increased. This indicates the formation of impurities on the surface of the framework after modification by magnetron sputtering. Therefore, the carbon peaks observed in the survey XPS spectra (Appendix A) of the modified PLGA scaffolds indicated not only polymer chemical bonds, but also indicated a carbon contamination that is most likely originated from hydrocarbon in the air. Such contaminations were present in the ambient air [63]. In reference [63], it is shown that the carbon contaminations are deposited on the surface of silver substrates exposed to the ambient air, with an increase in the concentration of carbon from 0% to 24% within 4 days.

The C1s, O1s, Cu2p3/2 and Ti2p core level spectra of unmodified and surface-modified PLGA scaffold samples are displayed in Figure 3.

Deconvolution of C1s spectrum of the unmodified PLGA scaffold sample (u) allowed the identification of three carbonaceous compounds, represented by the peaks with the colors pink, brown and gray in the spectrum (Figure 3, left column). The pink colored fractions of the deconvoluted of the C1s spectra were in the binding energy 284.8 eV and represent the C–C/C–H bonds of the polymer PLGA. The brown colored part of the deconvoluted C1s spectrum that were in the binding energy of 286.8 eV represents the C–O compounds of PLGA. The gray colored proportion of the deconvoluted C1s spectra were in the binding energy of 288.9 eV and refer to the C=O bonds of the polymer [64,65,66]. The shape, intensity and position of each component corresponded to the PLGA polymer [67].

In each XPS C1s spectrum of the Cu-Ti modified PLGA samples (LCu-Ti, MCu-Ti, HCu-Ti), three carbon compounds are present and illustrated by the colors pink, brown and gray (Figure 3, left column). The pink colored peaks were located at a binding energy of 285.2 eV, indicating C-C/C-H chemical bonds. Brown colored peaks at binding energies in the range of 286.4–286.5 eV indicated C-O bonds, and gray peaks at binding energies in the range of 288.7–288.8 eV indicated C=O bonds. After surface modification of the PLGA scaffolds, the C-C/C-H and C-O bonds shifted 0.4 eV to the left and right, respectively. This energy shift of the carbon chemical bonds could be related to the fact that the C1s spectra of the modified PLGA scaffolds demonstrated not only peaks that were indicative of a polymer-derived part of the carbon, but also an organic contamination. The position of the compounds and their peak intensities were in agreement with a work in which organic contaminations on metal substrates were studied [68]. Similar peak positions and intensities of contaminations with organic compounds were also shown in reference [69], in which composite coatings were deposited on the surface of fibrous cotton materials by magnetron co-sputtering. Moreover, such a shift of the carbon peaks after magnetron sputtering indicated a charging effect in the metal deposition process [70].

In the Cu2p_3/2_ spectra of the surface-modified PLGA samples (LCu-Ti, MCu-Ti, HCu-Ti), high-intensity Cu2p_3/2_ peaks and broad low-intensity Cu2p_3/2_ satellite peaks were present (Sat Cu2p_3/2_, Figure 3). These satellite peaks, which were located at higher binging energies at the main XPS peaks, indicated the typical shake-up effect of copper [71]. Deconvolution of the Cu2p_3/2_ peaks could distinguish three compounds corresponding to different Cu bonds. The red, green and blue components at binding energies in the range of 932.1–932.3 eV, 933.6–933.7 eV and 934.8–934.9 eV refer to the copper compounds Cu/Cu_2_O, CuO and Cu(OH)_2_ (Figure 3, column in the middle) [71]. PLGA scaffold samples HCu-Ti and LCu-Ti had the highest and lowest intensities of copper-containing chemical compounds, respectively (Figure 3, column in the middle). The Cu2p_3/2_ satellite peaks, which ranged between ~940 and 945 eV arose from the shake-up effect, increasing with the amount of copper content in the Cu-Ti alloy thin film on the surface of the modified PLGA scaffolds.

Two high intensity peaks corresponding to Ti2p_3/2_ and Ti2p_1/2_ were present in the Ti2p core spectra of the surface-modified PLGA scaffold samples. Deconvolution of the Ti2p core spectra allowed the differentiation of six components in each sample, representing different titanium compounds (Figure 3, right column). The most intensive purple-blue peaks at binding energies in the range of 458.8–458.9 eV and 464.5–464.6 eV indicated the presence of TiO_2_, dark cyan peaks at 454.6–454.7 eV and at 460.4–460.5 eV corresponded to TiO and the orange peaks at 462.0–462.1 eV and 456.5–456.6 eV corresponded to Ti_2_O_3_ [72,73,74]. Moreover, the surface-modified sample LCu-Ti had the most intense Ti2p peak, the sample MCu-Ti had the lowest peak intensity of all Ti2p spectra (Figure 3).

After the surface modification of the PLGA scaffold samples with Cu and Ti by magnetron co-sputtering, metallic composite compounds containing various copper oxides (CuO/Cu_2_O/Cu(OH)_2_) and titanium oxides (TiO/TiO_2_/Ti_2_O_3_) in different concentrations were formed. The appearance of such compounds was due to the fact that the chemically active plasma-modified surface of PLGA scaffolds can also actively interact with oxygen in the ambient air [75]. Qin et al. demonstrated that Cu-Ti composite thin films fabricated by magnetron sputtering have TiO, Ti_2_O_3_, TiO_2_, Cu_2_O, and CuO compounds [53]. The formation of metal-oxygen species was most likely caused by the chemical bonding of deposited Cu, Ti atoms with O_2_ taken from ambient air for thicker film thicknesses, whereby for the first layer the Cu and Ti atoms are also interacting chemically with the functional groups of the polymer backbone. The formation of copper(II) hydroxide (Cu(OH)_2_) on the surface of PLGA scaffolds was most likely caused by the interaction of plasma activated copper with water from the atmosphere [76].

### 3.3. Morphology and Elemental Mapping of Sample Cross-Sections

The EDX mapping of unmodified PLGA scaffolds reveals no Cu and Ti elements (Appendix A). The EDX mappings of surface-modified PLGA scaffolds displayed Cu and Ti, indicating that the surfaces of these scaffolds were completely covered by Cu-Ti alloy thin film coatings. Within the bulk volume of the surface-modified PLGA scaffolds, Cu and Ti elements were almost absent. This indicates that the surface modification performed by magnetron co-sputtering only coated the near-surface layers of the polymer scaffolds, while the internal bulk volume remained unmodified. The predominant modification of the near-surface layers of polymer scaffolds was associated with the shading effect [77], whereby the near-surface PLGA fibers prevented the coating of fibers lying deeper in the polymer scaffolds.

### 3.4. Wettability and Mechanical Properties

Water contact angle (WCA) of unmodified PLGA scaffolds is 121 ± 9° (Appendix A). The WCA of Cu-Ti modified PLGA samples were in the range of (125–128) ± 11° (Appendix A). After surface modification of the PLGA scaffolds by deposition of Cu-Ti alloy thin films, the WCA remained almost the same as for the unmodified scaffold. The Cu/Ti ratio of the thin film on the surfaces of modified PLGA scaffolds did not affect the wettability significantly.

The maintaining of the wettability for all PLGA scaffold samples, even after Cu-Ti modification, is associated with the following points:(1)Preservation of the surface morphology and surface roughness of PLGA scaffolds also after their Cu-Ti modification, which are reflected in the results of SEM and AFM (Figure 2).(2)Copper thin films, which are deposited on polymer substrates by magnetron sputtering, have a hydrophobic nature [78,79]. However, titanium thin films impart hydrophilic properties to polymer surfaces [80]. Thus, it can be concluded that even in the presence of copper on the surface of Cu-Ti-modified PLGA scaffolds in relatively low concentrations (Appendix A) is sufficient to preserve the hydrophobic properties.(3)Organic contaminations that are deposited on the modified scaffolds by a chemically active surface, adversely affect the wetting properties [81].

The tensile strength and maximum elongation at break of unmodified and surface-modified PLGA scaffold samples were in the range of (3.4–3.7) ± 0.3 MPa and (476–532) ± 67%, respectively (Appendix A). Surface modification by magnetron co-sputtering did not significantly affect the tensile strength and maximum elongation at break of PLGA scaffolds. The preservation of mechanical properties after surface modification of PLGA scaffolds by the applied method may be related to the modification of only superficial fibers, while the PLGA scaffolds remained intact within the volume.

### 3.5. Thermal Gravimetric Analysis (TGA)

The TGA and differential thermogravimetric (DTG) curves are shown in Appendix A. The average weight of the unmodified PLGA scaffold samples (u) was 3.22 ± 0.78 mg (Appendix A). The average weights of the surface-modified PLGA scaffold samples LCu-Ti, MCu-Ti, HCu-Ti were 3.83 ± 0.91 mg, 3.81 ± 1.21 mg and 3.54 ± 1.02 mg, respectively (Appendix A). After thermal gravimetric analysis, the unmodified PLGA scaffold samples were completely burned, while the surface-modified scaffold samples LCu-Ti, MCu-Ti, HCu-Ti left unburned rests with the following masses: 0.31 ± 0.24 mg (8.2 ± 7.5%), 0.27 ± 0.18 mg (7.1 ± 4.7%) and 0.23 ± 0.21 mg (6.4 ± 5.0%), respectively (Appendix A). This indicates that titanium and copper films were deposited on the surface of the PLGA fibers, with relative weights in the range of (6.4–8.2) ± 7.5% of the total weight of the surface-modified scaffolds. The weight of the metal films on the surface of the surface-modified PLGA scaffolds did not change reliably and did not depend on the applied operating parameters of the surface modification mode (Appendix A). This was because the total amount of copper and titanium was between 15.0% to 15.5% on the surface of the surface-modified PLGA scaffolds (see Section 3.2).

The thermal decomposition temperature of the unmodified PLGA scaffold samples was 309 °C (Appendix A). For the surface-modified PLGA scaffold samples, the thermal decomposition temperatures were the range of 303–305 °C. This leads to the conclusion that the surface modification of PLGA scaffolds by magnetron sputtering with Cu and Ti did not significantly change the thermal decomposition temperature.

### 3.6. Copper Ion Release, Elemental Mapping, Antibacterial Activity

Released amount of copper ions, release rate of copper ions and antibacterial activity of unmodified and surface-modified PLGA scaffold samples are presented in Figure 4.

The lowest and highest amount of copper ions and copper release rates were observed for the LCu-Ti and HCu-Ti samples, respectively (Figure 4a,b and Appendix A).

For surface-modified PLGA sample with the lowest Cu/Ti ratio LCu-Ti, the maximum amount of copper ions is 0.10 ± 0.03 mg∙L^−1^ determined on day 14 (Figure 4a and Appendix A). For the MCu-Ti sample with an average Cu/Ti ratio, the maximum amount of copper ions is 1.59 ± 0.48 mg∙L^−1^ measured on day 28 (Figure 4a and Appendix A). Finally, for the HCu-Ti sample with a highest Cu/Ti ratio, the maximum amount of copper ions was 2.41 ± 0.13 mg∙L^−1^ observed on day 14 (Figure 4a and Appendix A). The maximum amount of copper ions for HCu-Ti sample was 1.5 and 24 times higher, than the amount of copper ions for MCu-Ti and LCu-Ti samples, respectively. For LCu-Ti and HCu-Ti samples, the maximum amounts of copper ions were defined after 14 days from placing the samples in a test tube with deionized water. This indicates that after 14 days the amount of copper in the solution with LCu-Ti and HCu-Ti samples reached the maximum.

For the surface-modified PLGA sample with the lowest Cu/Ti ratio LCu-Ti, the release rate of copper ions after 30 min (day 0) is 0.96 ± 0.20 mg∙L^−1^∙d^−1^ (Figure 4b and Appendix A), on the days 1 and 3, the copper release rate was 0.02 ± 0.00 mg∙L^−1^∙d^−1^ and 0.01 ± 0.00 mg∙L^−1^∙d^−1^, and on the days 7, 14 and 28, the copper release rate dropped to 0 mg∙L^−1^∙d^−1^. The release rate for the MCu-Ti sample after 30 min (day 0) was 2.02 ± 0.82 mg∙L^−1^∙d^−1^ (Figure 4b and Appendix A), on the days 1 and 3, the copper release rate was 0.12 ± 0.04 mg∙L^−1^∙d^−1^ and 0.25 ± 0.08 mg∙L^−1^∙d^−1^, on the days 7 and 14, the amount of the released copper was 0.10 ± 0.03 mg∙L^−1^∙d^−1^ and 0.01 ± 0.00 mg∙L^−1^∙d^−1^, and on day 28, the amount of released copper decreased to 0 mg∙L^−1^∙d^−1^. For the HCu-Ti sample with the highest Cu/Ti ratio, the release rate after 30 min (day 0) was 2.69 ± 0.62 mg∙L^−1^∙d^−1^ (Figure 4b and Appendix A), on the days 1 and 3, the release rate of copper ions was 0.47 ± 0.14 mg∙L^−1^∙d^−1^ and 0.06 ± 0.02 mg∙L^−1^∙d^−1^, on the days 7 and 14, the amount of the released copper was 0.22 ± 0.06 mg∙L^−1^∙d^−1^ and 0.13 ± 0.04 mg∙L^−1^∙d^−1^, and on day 28, the release rate of copper dropped also for this sample to 0 mg∙L^−1^∙d^−1^. With increasing time, the release rate of the copper ions from the surface-modified PLGA scaffolds in deionized water decreased significantly (Figure 4b and Appendix A). The HCu-Ti sample had ~1.33 and ~2.8 times higher release rate of copper ions after 30 min than the MCu-Ti and LCu-Ti samples, respectively.

The maximum and minimum amount and release rate of copper ions were found for the samples with biggest Cu/Ti ratio (HCu-Ti) and lowest Cu/Ti ratio (LCu-Ti), respectively (Figure 4a,b and Appendix A). Therefore, a change in the Cu/Ti ratio of the films deposited on the surface of PLGA scaffolds significantly affected the amount and release rate of copper ions. The same trend has been reported in other studies [82,83], where antibacterial and biological investigations of Cu-Ti thin films were carried out.

As it can be seen on the EDX mapping micrographs of the surface-modified (LCu-Ti, MCu-Ti, HCu-Ti) PLGA scaffold samples, it was noticeable that copper (Cu) and titanium (Ti) were predominantly located on the surface of the distinct fibers (Appendix A). Oxygen (O) was also found on the surface of polymer fibers, but the intensity of this element was lower than intensity of Cu and Ti (Appendix A, third column). This finding confirmed that copper and titanium oxide coatings were formed on the surface of the fibers. Individual fibers could not be identified on the EDX mapping micrographs of carbon (C), because of a blurred spectral appearance in the micrographs (Appendix A, second column). This effect can be explained by the fact that the PLGA samples were glued with conductive tape on the specimen holder, which was predominantly composed of carbon. During the interaction of the surface-modified PLGA samples (LCu-Ti, MCu-Ti, HCu-Ti) with deionized water for 14 and 28 days, the intensity of oxygen, copper and titanium on the EDX mapping micrographs slightly decreased (Appendix A). This indicates that after 14 and 28 days, the metal films consisting of copper and titanium oxides on PLGA fibers were partially dissolved in water, which corresponded to the results for the copper release. After 14 and 28 days, the morphology of the PLGA scaffolds remained unchanged, indicating that the samples would not be degraded during this time (Appendix A, left column).

Unmodified and modified LCu-Ti samples showed no antibacterial properties (Figure 4c,d). During various incubation times, the number of bacteria in contact with these samples freely increased (Figure 4c), as evidenced by the near-zero values of the antibacterial activity indicator (0 to 14%) (Figure 4d). With increasing incubation time (from 6 to 24 h), the antibacterial activity indicator for MCu-Ti sample decreased from 61 ± 10% to 40 ± 8%, which indicates an increase in the number of MRSA colonies from 820 ± 140 CFU∙mL^−1^ to 1250 ± 240 CFU∙mL^−1^ (Figure 4c,d). For the HCu-Ti sample, with increasing incubation time (from 6 to 24 h), the antibacterial activity indicator decreased from 93 ± 13% to 62 ± 7%, which corresponds in the increase in MRSA colonies from 140 ± 20 CFU∙mL^−1^ to 790 ± 90 CFU∙mL^−1^ (Figure 4c,d). MRSA counts were made from Petri dishes shown in Appendix A. With increasing incubation time, the number of bacteria colonies increased, indicating that MCu-Ti and LCu-Ti samples have bacteriostatic properties. Bacteriostatic properties of materials means that they inhibit the growth of microorganisms [84]. The HCu-Ti PLGA showed the best antibacterial properties, because these samples had the highest indicator of antibacterial activity and therefore the lowest number of MRSA colonies in the experiment.

The enhancement of the antibacterial activity of the surface-modified PLGA samples depended directly on the concentration of copper in the thin film on the scaffold fiber surfaces and thus also on the amount of copper ions released over time. In this study, the highest level of released copper ions and antibacterial activity was observed for the HCu-Ti samples, which contained the highest Cu amount and the lowest Ti amount of all surface-modified PLGA scaffolds. The LCu-Ti samples did not show antibacterial properties, since the lowest amount of copper ions were released from this sample type (Figure 4a). Copper containing nanoparticles and thin films were known to have antibacterial properties [85,86,87]. An increase in the copper concentration in copper-titanium thin films increases their ability to inhibit pathogenic microorganisms [82]. At the same time, thin films, which mainly consist of titanium, have no antibacterial properties [82].

### 3.7. Toxicity

The toxicity of unmodified and surface-modified PLGA scaffold samples to NIH/3T3 cells and human gingival fibroblasts (HGFs) are presented in Figure 5.

Based on the results of optical density (OD) and relative growth rate (RGR), the unmodified PLGA scaffold samples (u) exhibited no toxic effect on NIH/3T3 cells (Figure 5a–d) and human gingival fibroblasts (Figure 5e,f).

With an increase in the incubation time of NIH/3T3 cells from 24 to 96 h, the OD increased by ~1.4–1.8 times and thus the amount of living cells (Figure 5d). This indicates beneficial conditions for the NIH/3T3 cell growth with the PLGA samples examined. According to the toxicity reaction classification criteria, a “Relative Growth Rate” (RGR) value above 75% is considered safe for cell growth [88]. The RGR values of NIH/3T3 cells for all modified PLGA scaffold samples (LCu-Ti, MCu-Ti, HCu-Ti) exceeded 75% (Figure 5b–d), indicating their non-toxicity to these cell types. With the increase in the Cu/Ti ratio, the RGR values for NIH/3T3 cells slightly decreased by 1.07 times for the LCu-Ti sample and 1.15 times for the HCu-Ti sample. With 10 times and 100 times dilution of the extracts from the Cu-Ti modified PLGA samples, the RGR values increased almost to the values obtained from the unmodified PLGA scaffold samples (88–120%) (Figure 5b–d). Therefore, in case of incorporation of a surface-modified PLGA scaffold sample into a living body, their toxicity will decline over time, since copper will be washed out from the thin film on the scaffold surface.

According to the results of the toxicity analysis of surface-modified PLGA scaffolds to human gingival fibroblasts, with an increase in the Cu/Ti ratio for the thin films on the PLGA samples, the OD and RGR values decreased significantly (Figure 5e,f). The LCu-Ti samples with the lowest Cu/Ti ratio have a RGR value, which is ~2 times higher than for the HCu-Ti samples, which had the biggest ratio of Cu/Ti. According to the toxicity reaction classification criteria, the HCu-Ti PLGA sample with a RGR value of 51 ± 5% was toxic to human gingival fibroblasts, LCu-Ti and MCu-Ti samples with a RGR value of 104 ± 13% and 76 ± 13% were non-toxic (Figure 5f).

According to the toxicity results, it can be seen that the toxic effect of Cu-Ti modified scaffolds increased with increasing copper content. However, all Cu-Ti modified sample extracts showed no toxicity to NIH/3T3 cells, and only the HCu-Ti scaffold samples with the highest Cu amount demonstrate toxicity to human gingival fibroblasts, which had contact directly with the surface of modified PLGA scaffolds (see Section 2.15). The increase in toxicity with an increase in the Cu amount on the surface of PLGA scaffolds was directly related with an increase in the release rate of copper ions from the samples. In conclusion, results obtained in this study correlate with the works of others [82,89]. Wojcieszak et al. reported that for Cu-Ti coatings, the amount of copper ions released and the toxicity towards mouse fibroblasts L929 increased with increasing Cu/Ti ratio [82].

Optical micrographs of NIH/3T3 cells at 500× magnification are shown in the supporting information (Appendix A), which were incubated with non-diluted PLGA sample extracts at different times (24, 48 and 96 h).

NIH/3T3 cells that were in contact with unmodified PLGA scaffold samples extracts showed a good cell proliferation behavior, according to the micrographs (u in Appendix A). The morphology of NIH/3T3 cells, which were in contact with Cu-Ti modified PLGA sample extracts (Appendix A, LCu-Ti, MCu-Ti, HCu-Ti) for various times (24, 48, 96 h) showed the same morphology as the cells in contact with unmodified samples extracts. Thus, the absence of toxicity of Cu-Ti modified PLGA scaffolds extracts for the NIH/3T3 cell line is confirmed.

The good morphology of NIH/3T3 cells during the experiments (Appendix A) and the absence of toxicity for surface-modified PLGA scaffolds (Figure 5a–d) is explained by the fact that such cells did not contact with PLGA scaffolds but interacted with copper ions released from the surface of PLGA scaffolds into the MEM solution (see Section 2.14). The release rate of copper ions from surface-modified PLGA scaffolds is not high enough to gain toxic effects to this cell type, which is proven by no negative effect on the number and morphology of NIH/3T3 cells in the experiments performed.

## 4. Conclusions

In present work, electrospun poly(lactide-co-glycolide) scaffolds were successfully copper-titanium modified by magnetron co-sputtering method in direct current pulsed mode. Antimicrobial activity and toxicity increase with increasing copper content, while titanium has a positive effect on biocompatibility and does not change the antibacterial properties of modified poly(lactide-co-glycolide) scaffolds. Magnetron plasma modification preserves the original morphology, mechanical properties and wettability of the unmodified poly(lactide-co-glycolide) scaffold. With the increase in the magnetron discharge power, the surface roughness of copper-titanium modified poly(lactide-co-glycolide) scaffolds increases for individual fibers. During the modification process, metallic thin films are formed on the surface of poly(lactide-co-glycolide) scaffold fibers consisting of copper oxide (CuO/Cu_2_O/Cu(OH)_2_) and titanium (TiO/TiO_2_/Ti_2_O_3_) compounds. Copper-titanium modified poly(lactide-co-glycolide) scaffolds with the lowest amount of copper, have no antibacterial properties and no toxicity towards NIH/3T3 cells and human gingival fibroblasts. Surface-modified poly(lactide-co-glycolide) scaffolds with a medium and a high amount of copper, exhibit antibacterial properties. The medium copper amount samples are non-toxic towards both tested cell cultures and only the samples with a high amount of copper are toxic to human gingival fibroblasts. Therefore, the samples with a medium amount of copper can be recommended for future tissue engineering applications, since they have a copper amount that is not toxic to cells, but at the same time suppresses the growth of antibiotic-resistant bacteria. The obtained results provide insights into the effects of plasma modification with copper and titanium on the morphology, physicochemical, antibacterial and biological properties of PLGA scaffolds. A future continuation of this work will be in vivo studies for oral soft tissue engineering using surface-modified PLGA scaffolds modified with coatings of copper and titanium taken into account the biological results of this study.

## Figures and Tables

**Figure 1 pharmaceutics-15-00939-f001:**
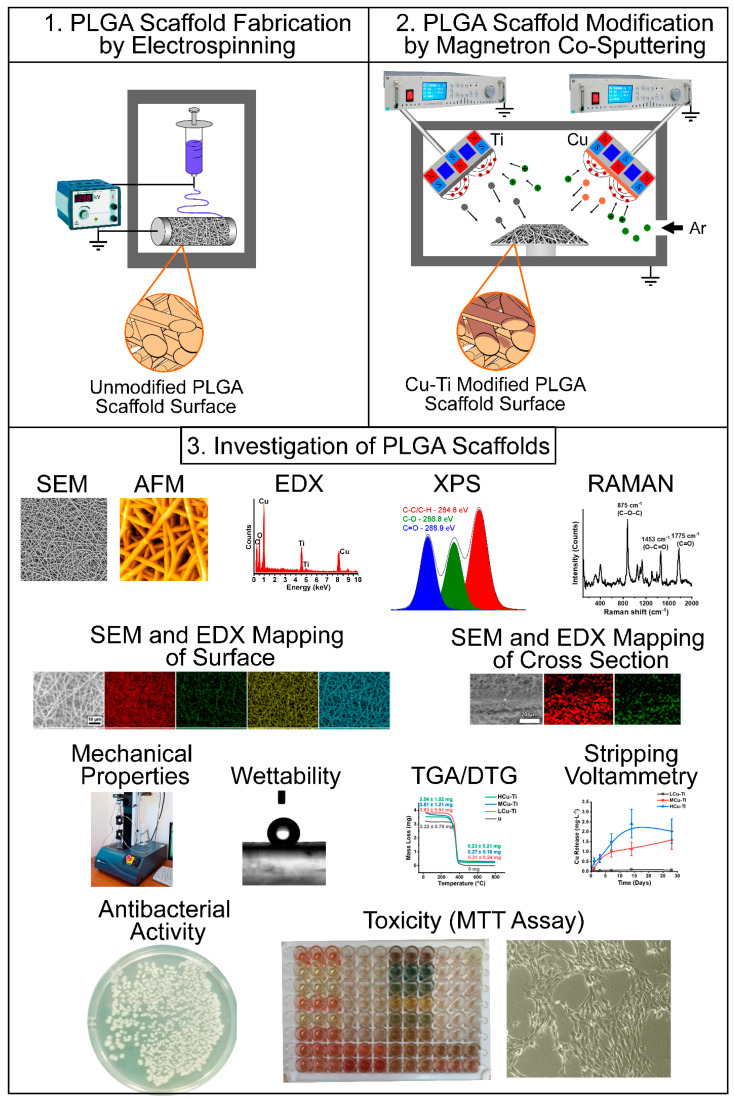
Schematic illustration of poly(lactide-co-glycolide) (PLGA) scaffold fabrication and surface modification. In the first stage, PLGA scaffolds were formed by electrospinning and in the second stage, the PLGA scaffolds were surface-modified by pulsed DC magnetron co-sputtering of copper (Cu) and titanium (Ti) to obtain a Cu-Ti coating. The third section of this figure is given the overview of all conducted investigation methods in this study.

**Figure 2 pharmaceutics-15-00939-f002:**
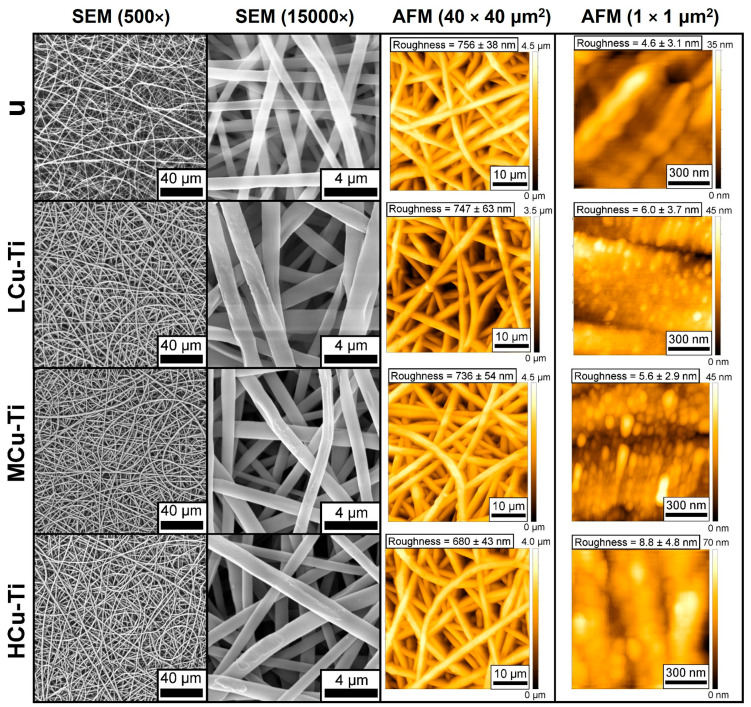
First column: SEM micrographs taken at 500× magnification; second column: SEM micrographs taken at 15,000× magnification; third column: AFM micrographs at 40 × 40 μm^2^ scanning area; fourth column: AFM micrographs at 1 × 1 μm^2^ scanning area of unmodified (u) and surface-modified (LCu-Ti, MCu-Ti and HCu-Ti) PLGA scaffold samples.

**Figure 3 pharmaceutics-15-00939-f003:**
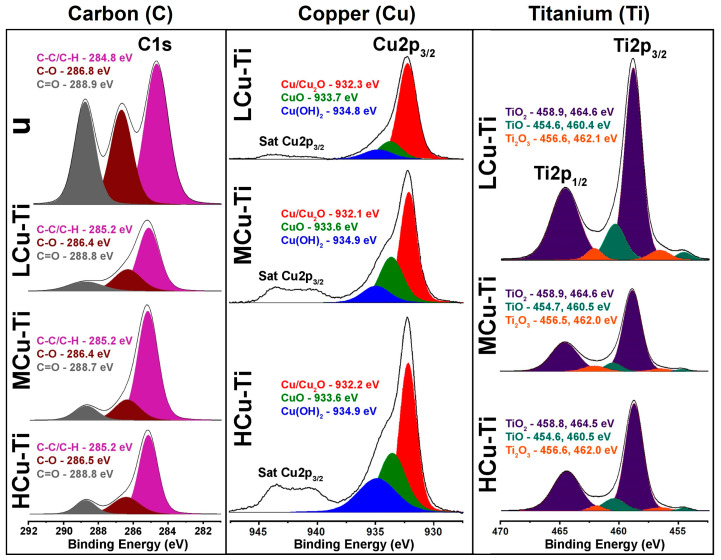
C1s, Cu2p3/2 and Ti2p XPS core spectra of unmodified (u) and surface-modified PLGA scaffold samples (LCu-Ti, MCu-Ti, HCu-Ti). Note: The unmodified PLGA scaffolds (u) contains no Cu and Ti and thus the XPS spectra shows no Cu2p_3/2_ and Ti2p_3/2_ peaks, because of not being surface-modified by co-sputtering of Cu and Ti.

**Figure 4 pharmaceutics-15-00939-f004:**
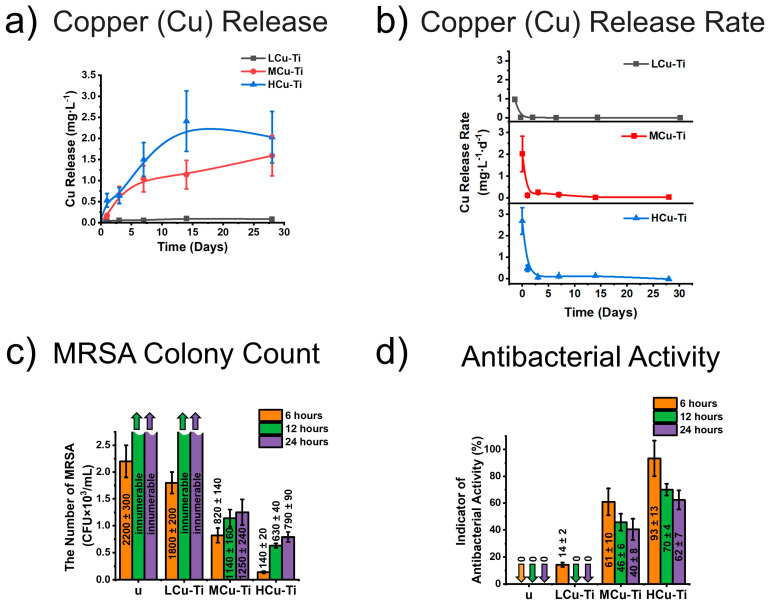
Released amount of copper ions and antibacterial activity of unmodified (u) and surface-modified PLGA scaffold samples (LCu-Ti, MCu-Ti, HCu-Ti): (**a**) released amount of copper ions and (**b**) release rate of copper ions into deionized water from Cu-Ti modified PLGA scaffolds during 28 days, determined via stripping voltammetry, (**c**) number of MRSA colonies (Appendix A), the word “innumerable” refers to that the Petri dish contained so many MRSA colonies that they could not be counted; (**d**) antibacterial activities calculated on the basis of the number of MRSA colonies in.

**Figure 5 pharmaceutics-15-00939-f005:**
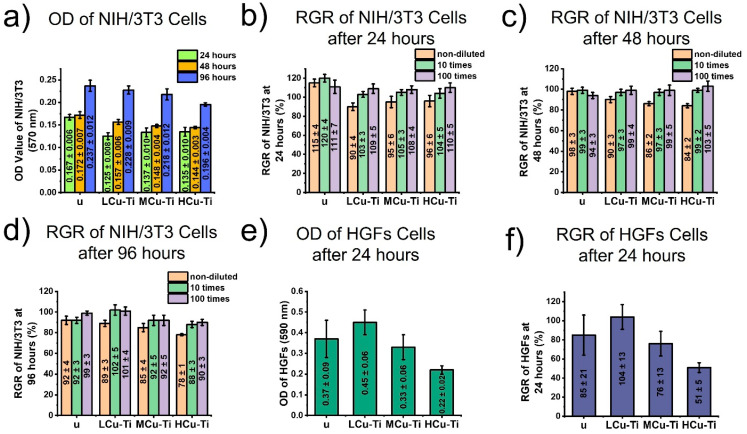
Cytotoxicity of unmodified (u) and surface-modified PLGA scaffold samples (LCu-Ti, MCu-Ti, HCu-Ti): (**a**) optical density (OD) of NIH/3T3 cells incubated with non-diluted PLGA sample extracted at different times (24, 48 and 96 h); (**b**–**d**) relative growth factors (RGR) of NIH/3T3 cells incubated with non-diluted, 10 times diluted and 100 times diluted PLGA sample extracted after 24 h (**b**), 48 h (**c**) and 96 h (**d**); (**e**) optical density (OD) of human gingival fibroblasts incubated with PLGA samples for 24 h; (**f**) relative growth rate (RGR) of human gingival fibroblasts incubated with PLGA samples for 24 h.

## Data Availability

Data underlying the results presented in this paper are not publicly available at this time but may be obtained from the authors upon reasonable request.

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
