# Peer review of "Antibacterial Activity and Cytocompatibility of Electrospun PLGA Scaffolds Surface-Modified by Pulsed DC Magnetron Co-Sputtering of Copper and Titanium"

_pharmaceutics, 2023, doi:10.3390/pharmaceutics15030939_

Round 1
Reviewer 1 Report
The article is well written and the work is interesting and elaborate. This article can be accepted after minor modifications based on the following comments:
1. The details inside some of the figures are not clear.
2. In the Introduction section, it is not clear why the authors have chosen synthetic polymer rather than natural polymer.
3. Why 4% PLGA for the scaffold fabrication? Any specific reasons
4. Future work based on this can be mentioned in conlusion or at the end of Discussion part.
Author Response
General answer: We like to thank the reviewer for the kind judgement of our work. All comments were considered during our revision. Changes in the manuscript have been marked in green.
- The details inside some of the figures are not clear.
Answer 1: We thank the reviewer for this observation. We made changes to figures 2, 3, 4 and 5 and hope that these changes will contribute to a better understanding of these figures.
- In the Introduction section, it is not clear why the authors have chosen synthetic polymer rather than natural polymer.
Answer 2: We thank the reviewer for pointing out this shortcoming. In accordance with this comment, we have added the as few sentences to the introduction in line 43 – 47:
Among the most common natural polymers used in tissue engineering are collagen and gelatin [9]. Despite the high biocompatibility of collagen and gelatin, they have low mechanical strength [10,11] and a very high rate of degradation. The degradation of collagen is up to 4 weeks in vivo and a complete degradation of gelatin up to 2 – 4 weeks [12,13].
The reasons why we chose PLGA are described in the introduction in lines 47 – 53:
Compared to PCL, PLA, collagen and gelatin, PLGA has a controlled degradation rate that depends on the ratio of lactide/glycolide for synthesis [14]. PLGA has been approved by the US Food and Drug Administration for clinical use [15]. This happened because PLGA has good biocompatibility and mechanical properties that are suitable for applications in tissue engineering [16]. PLGA is also of high interest because of its use for drug delivery applications from a biodegradable polymer matrix [17].
- Why 4% PLGA for the scaffold fabrication? Any specific reasons
Answer 3: We thank the reviewer for this question. To answer this question: We have done preliminary studies that have shown that different solution concentrations (2%, 3%, 4%, 5%) yield different morphologies of PLGA scaffolds. Based on the preliminary experiments performed, we found that the PLGA scaffolds prepared from a 4% PLGA solution had the best morphology and mechanical properties for use in tissue engineering. The manuscript for this paper is already in preparation, but some investigation results are still missing. We plan to submit this paper for publication in the summer of this year.
- Future work based on this can be mentioned in conlusion or at the end of Discussion part:
Answer 4: We thank the reviewer for this suggestion. We have added an outlook on the planned further studies in the conclusion.
Reviewer 2 Report
The authors describe electrospun PLGA scaffolds surface-modified by pulsed DC magnetron co-sputtering of copper and titanium. The topic is interesting and provides a proper technological solution to specific problems of bioresorbable implants for the regeneration of fast-growing tissues. Overall, the quality of the manuscript is very good. It is well-written and easy to read. The introduction provides a sufficient background of the topic with a clear aim setting. No additional measurements are required to reach the conclusions.
Specific comments:
The authors must check for any grammar or spelling mistakes, eg. :
Line 25 „scaffold samples (…) shows”
Line 507: „scaffolds remains”
Author Response
We would like to thank the reviewer for this kind assessment of our work. The manuscript has been corrected according to the comments: Line 27 to 29 „scaffold samples (…) shows” has been changed to „scaffold samples (…) show” and line 510 (new): „scaffolds remains” has been changed to „scaffolds remain”. All changes are marked in green in the manuscript.
Reviewer 3 Report
After reading the manuscript, I found the topic truly interesting, well-written, with high scientific soundness, abstract and introduction successfully give and explain the necessary information to the reader to understand the topic, some details are missing in methodology, but in general is well explained, results are logical and well discussed. Graphics and figures are excellent, self explained and esthetics, congratulations. Conclusion is well structured and references are quite and pertinent to the topic. Just a minimal adjusting comments are given below to improve the quality of the manuscript
Suggestions
a) Please add a material section after section 2, to enlist all polymers (adding MW and brand), solvents (brand and purity), and biological cell lines (ATTC number).
b) for the electrospinning conditions, please add the temperature and relative % humidity used in the process
c) How many replicates have you used in mechanical testing, antimicrobial, and cytotoxicity studies?, is not explained in the methodology
d) Please put in italics format all biological species names, and Latin words such as "in vitro" and "in vivo"
Author Response
General answer: We like to thank the reviewer for the kind judgement and appreciation of our work. The given suggestions were taken into account in the revision. All changes in the manuscript have been marked in green.
Suggestions
- a) Please add a material section after section 2, to enlist all polymers (adding MW and brand), solvents (brand and purity), and biological cell lines (ATTC number).
Answer: We like to thank the reviewer for this suggestion. The required section has been added to chapter 2 as chapter 2.1 Materials. All other chapters have been renumbered.
- b) for the electrospinning conditions, please add the temperature and relative % humidity used in the process
Answer: We thank the reviewer for this helpful comment. The required information was added in chapter 2.2 on page 3.
- c) How many replicates have you used in mechanical testing, antimicrobial, and cytotoxicity studies?, is not explained in the methodology
Answer: We thank the reviewer for pointing out this missing information. The required information was added in chapter 2.9 on page 6, in chapter 2.14 and 2.16 on page 7.
- d) Please put in italics format all biological species names, and Latin words such as "in vitro" and "in vivo"
Answer: We thank the reviewer for this useful suggestion. The terms mentioned have been changed to italics.
Reviewer 4 Report
Interaction of scaffold surface with cells is one of the most important issues within tissue engineering. Affinity of living cells to the unmodified PLGA polymers was previously studied by some teams and seems to be relatively low. However, various cells are able to attach and grow on scaffolds made from PLGA. Surface modification of such scaffolds requires an extensive evaluation of potential chnges in cell adhesion, spreading, proliferation and migration within scaffolds. Assays done by the authors do not provide sufficient information of that type. Described surface modifications may turn out completely useless if cells are unable to grow on such structures.
Methods used in cytotoxicity studies are described very poorly and large amount of information is lacking. I will indicate only some of them:
- there is no any mention about the weight of extracted specimens and volume of extraction medium
- absurd sentence in L283 and L284 - how the authors ensured the "final" cell number being 10^4?
- how many technical and biological replicates were applied?
- Why the Mann-Whitney U test was used? It enables comparison only two groups. If there are numerous groups other tests should be used (as Kruskal-Wallis test).
- There is no any information about Mn, Mw or microstructure of PLGA.
I am unable to describe all shortcomings of the article but it should be definitely rejected.
Author Response
General answer: We thank the reviewer for reviewing our work. The biological tests we performed in agreement with international standards for validating the biocompatibility of medical implants in agreement with the registration procedures of the European Union, which all follow EN ISO 1009-1 and EN ISO 10993-5. We clarified this point with our German partner who is a member of the ISO committee ISO-TC 150-JWG 1 for the evaluation and registration of biomedical products, especially implants.
Methods used in cytotoxicity studies are described very poorly and large amount of information is lacking. I will indicate only some of them:
- there is no any mention about the weight of extracted specimens and volume of extraction medium
Answer: We thank the reviewer for this helpful comment. The weight of the samples with an area of 1 cm2 is about 3 – 4 mg. This information was already available in chapter 2.11 at line 230. In Chapter 2.15, we added the information about the surface-to-volume ratio of the extraction medium in lines 297 – 298.
- absurd sentence in L283 and L284 - how the authors ensured the "final" cell number being 10^4?
Answer: We thank the reviewer for pointing out this issue. This information has been added and clarified in chapter 2.15 in lines 303 – 307.
- how many technical and biological replicates were applied?
Answer: We thank the reviewer for pointing out this missing information. The required information was added in chapter 2.15 (lines 299 – 300) and in chapter 2.16 (lines 321 – 322).
- Why the Mann-Whitney U test was used? It enables comparison only two groups. If there are numerous groups other tests should be used (as Kruskal-Wallis test).
Answer: We thank the reviewer for this question. To answer this question: We applied the nonparametric Mann-Whitney test because the samples contained few values (mechanical properties, antibacterial properties, toxicity, porosity) or the samples did not conform to a normal distribution (value of fiber diameter or pore areas). The Kruskal-Wallis test performed gave the same results as the Mann-Whitney test, so we focused on the Mann-Whitney test.
- There is no any information about Mn, Mw or microstructure of PLGA.
Answer: We thank the reviewer for this comment. The information about Mn and Mw of PLGA used has been added to chapter 2.1 in line 109.
I am unable to describe all shortcomings of the article but it should be definitely rejected.
Answer: We are concerned about this unclear statement. Please clarify any other issues so that we can address them to improve the quality of our manuscript.
Reviewer 5 Report
This manuscript investigated the surface modification of biocompatible poly(lactide-co-glycolide) scaffolds in order to improve antibacterial properties of this type of scaffolds, as it can higher their application possibilities in medicine. Therefore, the scaffolds were surface-modified by means of pulsed direct current magnetron co-sputtering of copper and titanium targets in an inert atmosphere of argon. The results shows that the scaffold samples surface-modified with the highest copper to titanium ratio shows the best antibacterial properties and no toxicity against mouse fibroblasts, but have a toxic effect to human gingival fibroblasts. The optimal poly(lactide-co-glycolide) scaffold sample is surface-modified with a medium ratio of copper and titanium that has antibacterial properties and is non-toxic to both cell cultures. The data in this work can support the conclusion, and this manuscript can meet the scope of the journal of “Pharmaceutics”. However, there are still some problems in this manuscript. It needs a minor revision before becomes acceptable for publication in this journal.
1. Page 2, What is the molecule weight of PLGA?
2. Page 9, As shown in Figure S1, the porosity values of surface-modified by copper-titanium (Cu-Ti) thin films with different copper amounts (LCu-Ti, MCu-Ti, HCu-Ti) scaffolds are higher than that of unmodified scaffold. What is the reason?
3. In this manuscript, there are several mistakes. For example:
(1) Page 13, “The Cu/Ti ratio of the thin film on the surfaces of modified PLGA scaffolds does not effect on the wettability.”
(2) Page 18, “Wojcieszak et al. reported that for Cu-Ti coatings the amount of copper ions released and the toxicity towards mouse fibroblasts L929 increased with increasing Cu/Ti ratio [77]].”
Author Response
General answer: We like to thank the reviewer for the judgement and appreciation of our work. All changes in the manuscript have been marked in green.
- Page 2, What is the molecule weight of PLGA?
Answer: We thank the reviewer for this question. The information about the molecular weight of the PLGA used has been added to chapter 2.1 on page 3.
- Page 9, As shown in Figure S1, the porosity values of surface-modified by copper-titanium (Cu-Ti) thin films with different copper amounts (LCu-Ti, MCu-Ti, HCu-Ti) scaffolds are higher than that of unmodified scaffold. What is the reason?
Answer: We would like to thank the reviewer for this question. We observed that the porosity did not change significantly when the scaffolds were modified with PLGA. Although it can be seen that the porosity of unmodified fibers is slightly lower than that of surface-modified scaffolds, this difference is not statistically significant. In the manuscript, a sentence was added in lines 334 - 337 on page 9 to clarify this point.
- In this manuscript, there are several mistakes. For example:
(1) Page 13, “The Cu/Ti ratio of the thin film on the surfaces of modified PLGA scaffolds does not effect on the wettability.”
(2) Page 18, “Wojcieszak et al. reported that for Cu-Ti coatings the amount of copper ions released and the toxicity towards mouse fibroblasts L929 increased with increasing Cu/Ti ratio [77]].”
Answer: We would like to thank the reviewer for pointing out these mistakes:
- The word “significantly” has been added to this sentence.
- The doubled closing brackets at the end of the sentence has been removed.
Round 2
Reviewer 4 Report
The work has not been improved enough to be published